# Aquaporin-4 Expression during Toxic and Autoimmune Demyelination

**DOI:** 10.3390/cells9102187

**Published:** 2020-09-28

**Authors:** Sven Olaf Rohr, Theresa Greiner, Sarah Joost, Sandra Amor, Paul van der Valk, Christoph Schmitz, Markus Kipp

**Affiliations:** 1Institute of Anatomy, Rostock University Medical Center, Gertrudenstrasse 9, 18057 Rostock, Germany; sven.rohr@campus.lmu.de (S.O.R.); theresa.greiner@uni-rostock.de (T.G.); sarah.joost@med.uni-rostock.de (S.J.); 2Institute of Anatomy II, Faculty of Medicine, LMU Munich, Pettenkoferstrasse 11, 80336 Munich, Germany; christoph_schmitz@med.uni-muenchen.de; 3Department of Pathology, VU Medical Center, De Boelelaan 1117, ZH 3E36 1081 HV Amsterdam, The Netherlands; s.amor@amsterdamumc.nl (S.A.); p.vandervalk@amsterdamumc.nl (P.v.d.V.); 4Blizard Institute, Barts and The London School of Medicine and Dentistry, Queen Mary University of London, 4 Newark St., London E1 2AT, UK; 5Center for Transdisciplinary Neurosciences Rostock (CTNR), Rostock University Medical Center, Gelsheimer Strasse 20, 18147 Rostock, Germany

**Keywords:** aquaporin 4, cuprizone, experimental autoimmune encephalomyelitis, demyelination, blood–brain barrier, multiple sclerosis, neuromyelitis optica, digital histology

## Abstract

The water channel protein aquaporin-4 (AQP4) is required for a normal rate of water exchange across the blood–brain interface. Following the discovery that AQP4 is a possible autoantigen in neuromyelitis optica, the function of AQP4 in health and disease has become a research focus. While several studies have addressed the expression and function of AQP4 during inflammatory demyelination, relatively little is known about its expression during non-autoimmune-mediated myelin damage. In this study, we used the toxin-induced demyelination model cuprizone as well as a combination of metabolic and autoimmune myelin injury (i.e., Cup/EAE) to investigate AQP4 pathology. We show that during toxin-induced demyelination, diffuse AQP4 expression increases, while polarized AQP4 expression at the astrocyte endfeet decreases. The diffuse increased expression of AQP4 was verified in chronic-active multiple sclerosis lesions. Around inflammatory brain lesions, AQP4 expression dramatically decreased, especially at sites where peripheral immune cells penetrate the brain parenchyma. Humoral immune responses appear not to be involved in this process since no anti-AQP4 antibodies were detected in the serum of the experimental mice. We provide strong evidence that the diffuse increase in anti-AQP4 staining intensity is due to a metabolic injury to the brain, whereas the focal, perivascular loss of anti-AQP4 immunoreactivity is mediated by peripheral immune cells.

## 1. Introduction

Multiple sclerosis (MS) is widely considered to be an autoimmune, demyelinating disease of the central nervous system (CNS). Histopathological characteristics are demyelination, peripheral immune cell infiltration, gliosis, and axonal damage [1,2], which creates a neurodegenerative environment that leads to the accumulation of significant motor, sensory, and cognitive disabilities. The disease affects ~2,500,000 patients worldwide and is more common in females than in males. It is widely assumed that adaptive immunity generates lesions following the entry of leucocytes into the CNS [3]. These are associated with blood–brain barrier dysfunction, oligodendrocyte death, and the generation of demyelinated lesions within the white and grey matter. Consequently, during the last decades, the autoimmune aspects of the MS pathology have been in the focus of research efforts, in particular T- and, more recently, B-cell-mediated tissue injury. In the past years, however, increasing attention has been paid to the role of glia cells, in particular astrocytes, for MS lesion development and progression [4,5,6,7].

Astrocytes, the most abundant glia cell population in the CNS, are well known to play critical roles in CNS development, homeostasis, and response to injury. In addition to well-defined functions in synaptic signaling and blood–brain barrier control, astrocytes are now emerging as important contributors to white matter health [8,9]. Our group demonstrated that activation of the astrocytic anti-oxidative response ameliorates the formation of demyelinating lesions in a toxic MS animal model [10]. Recently, two distinct phenotypes of astrocytes, termed A1 and A2, with opposing functions have been identified [11]. While A1 astrocytes lose neuroprotective abilities and promote the death of oligodendrocytes and neurons, A2 astrocytes facilitate neuronal survival and tissue repair. Although the underlying mechanisms and cell-cell communication pathways of astrocyte-mediated tissue injury and repair remain to be clarified in MS and its animal models, astrocytes are direct targets in neuromyelitis optica (NMO). Neuromyelitis optica spectrum disorders (NMOSD), as currently defined, consist of a heterogeneous group of diseases characterized by neuroinflammation and demyelination that produce motor, visual, and other neurological deficits [12,13]. The inflammatory lesions mainly involve the optic nerve and/or spinal cord. Many NMOSD patients are seropositive for IgG autoantibodies against the astrocyte water channel aquaporin-4 (AQP4), called AQP4-IgG (or NMO-IgG). Pathological features of active NMO include astrocytopathy with loss of AQP4 and glia fibrillary acidic protein (GFAP), inflammation with granulocyte and macrophage infiltration, and microglial activation, activated complement deposition, blood–brain barrier disruption, and demyelination [14,15].

AQP4, first described in 1994 [16], is a water-selective transport protein abundantly expressed in glial cells bordering the subarachnoidal space, ventricles, and blood vessels [17]. AQP4 is also abundant in osmosensory areas, including the supraoptic nucleus and subfornical organ. Immunogold analyses demonstrated that AQP4 is restricted to glial membranes and subpopulations of ependymal cells. AQP4 is strongly expressed in glial membranes that are in direct contact with capillaries and the pia [17]. On the functional level, AQP4 can ameliorate brain edema after CNS injury [18,19], can regulate water transport in astrocytes [20,21], orchestrate astrocyte cell growth [21] or facilitate the reabsorption of excess fluid in vasogenic brain edema [22]. Furthermore, AQP4 appears to be important for acoustic signal transduction by facilitating rapid osmotic equilibration in epithelial cells in the organ of Corti [23].

Several animal models are used to study MS lesion development and progression. Experimental autoimmune encephalomyelitis (EAE) is the most commonly used tool to study the autoimmune aspects of MS. In EAE, experimental animals (commonly rodents) are immunized with a myelin-related peptide or protein antigen administered in a strong adjuvant, usually complete Freund’s adjuvant. Another model to study mechanisms and consequences of demyelination is the cuprizone model. In this model, cuprizone-induced metabolic injury results in oligodendrocyte stress [24] followed by massive primary oligodendrocyte apoptosis. This primary insult of oligodendrocytes leads to the activation of microglia and astrocytes, demyelination, and diffuse axonal injury [2,25,26,27,28]. We recently demonstrated that combining cuprizone-induced metabolic oligodendrocyte injury with active EAE induction results in intense, multifocal peripheral immune cell recruitment into the forebrain [29,30].

To understand the relevance of AQP4 during the development and progression of CNS injuries, further studies are needed addressing alterations in AQP4 expression and subcellular distribution related to specific pathologies. The purpose of this study was, therefore, to characterize in detail AQP4 expression in a model of metabolic (i.e., cuprizone) and autoimmune-mediated (i.e., combination of cuprizone and active EAE) CNS injury.

## 2. Materials and Methods

### 2.1. Animals and Groupings

All in vivo experiments were performed as published previously with minor modifications [29,30,31]. Eight-week-old C57BL/6 female and male mice (19 g–20 g) were purchased from Janvier Labs, Le Genest-Saint-Isle, France. hGFAP/EGFP transgenic mice [32] were used to visualize entire astrocyte cell bodies and processes. Mice were allowed to accommodate to the environment for at least 1 week before the beginning of the experiment. Microbiological monitoring was performed according to the Federation of European Laboratory Animal Science Associations recommendations. A maximum of five animals were housed per cage (cage area 435 cm^2^). Animals were kept at a 13 h light/11 h dark cycle, controlled temperature 22 °C ± 2 °C and 50% ± 10% humidity, with access to food and water *ad libitum*. It was assured that no light was used during the night cycle period. Nestlets were used for environmental enrichment. All experiments were formally approved by the Regierung Oberbayern (reference number 55.2-154-2532-73-15).

Mice were randomly assigned to the following experimental groups: (A) control (co), the animals were provided a diet of standard rodent chow for the entire duration of the study; (B) cuprizone, animals were intoxicated with a diet containing 0.25% cuprizone (Sigma-Aldrich, Taufkirchen, Germany) mixed into ground standard rodent chow for the indicated period (i.e., up to five weeks); (C) EAE, animals received the standard rodent chow for the entire duration of the study but were immunized with MOG_35–55_ at the beginning of week six; (D) Cup/EAE, mice were intoxicated with the cuprizone diet for the first three weeks, then received standard rodent chow and were then immunized with MOG_35–55_ at the beginning of week six as published previously [29,30].

### 2.2. Cuprizone-Induced Demyelination and Tissue Preparation

Cuprizone intoxication was performed as described previously [33,34]. In brief, the mice were fed a diet containing 0.25% cuprizone (Bis(cyclohexanone)-oxaldihydrazone, Sigma-Aldrich, Taufkirchen, Germany, catalog number C9012) mixed into ground standard rodent chow (Ssniff, Soest, Germany, catalog number V1530-0) for up to 5 weeks. Control mice were fed with cuprizone-free standard rodent chow in pellet formulation.

### 2.3. EAE and Disease Scoring

EAE induction and scoring were performed as published previously [29]. In brief, to induce the formation of encephalitogenic T cells, mice were immunized (s.c.) with an emulsion of MOG_35–55_ peptide dissolved in complete Freund’s adjuvant followed by injections of pertussis toxin in PBS (i.p.) on the day of, and the day after immunization (Hooke Laboratories, Inc., Lawrence, KS, USA). As outlined above, Cup/EAE mice were intoxicated with the cuprizone diet for the first three weeks and were then immunized with MOG_35–55_ at the beginning of week six.

The disease severity was scored as follows: A score of 1 was assigned if the entire tail dropped over the finger of the observer when the mouse was picked-up at the base of the tail; a score of 2 was assigned when the legs of the mice were not spread apart but held close together when the mouse was picked-up at the base of the tail, or when the mice exhibited a clearly apparent wobbly gait; a score of 3 was assigned when the tail was limp and the mice showed complete paralysis of hind legs (a score of 3.5 is given if the mouse is unable to raise itself when placed on its side); a score of 4 was assigned if the tail was limp and the mice showed complete hind leg and partial front leg paralysis, and the mouse was minimally moving around the cage but appears alert and feeding. A score of 4 was not attained by any of the mice in our study.

### 2.4. Tissue Preparation

At the end of the experiment, mice were deeply anesthetized and transcardially perfused with ice-cold phosphate-buffered saline (PBS) followed by a 3.7% formaldehyde solution (pH 7.4) or tissue dissection. For immunohistological studies, whole brains were dissected after overnight post-fixation in 3.7% formaldehyde solution and embedded in paraffin. Brains from hGFAP/EGFP transgenic mice were, after fixation, processed for cyroprotection and embedding in Tissue-Tek^®^ O.C.T.™ compound to preserve the EGFP fluorescence signal.

### 2.5. Human Samples

Paraffin-embedded post-mortem CNS tissues were obtained through a rapid autopsy protocol from subjects with progressive MS (in collaboration with The Netherlands brain bank Amsterdam). The rapid autopsy regimen of the Netherlands Brain Bank in Amsterdam (coordinator Dr I. Huitinga) was used to acquire the samples (granted in 2008, project number 938) with the approval of the Medical Ethical Committee of the Amsterdam UMC. All participants or next of kind had given informed consent for autopsy and use of their tissues for research purposes. 8 tissue blocks from 8 post-mortem brains (1 control) were included in the study. Patient details are shown in Table A1. The average age of patients was 58.85 years (mean) ± 14.3 years (standard deviation). Microscopic analyses of post-mortem samples was performed by two blinded readers (M.K. and P.v.d.V./S.A.). The staging of lesions was performed as reported previously [35].

### 2.6. Immunohistochemistry and Immunofluorescence

For immunohistological studies, coronal brain sections (5-μm-thick) were rehydrated and heat-unmasked in citrate-buffer (pH 6.0) or TRIS-EDTA-buffer (pH 9.0), if necessary. Before staining, sections were blocked in 5% serum in PBS (blocking solution; serum of species in which the secondary antibody was produced; Vector laboratories, Peterborough, UK) at room temperature for 1 h. Sections were then incubated with the primary antibodies diluted in blocking solution overnight at 4 °C. Appropriate negative controls included isotype antibodies if appropriate (equal concentration like the primary antibodies) and/or omitting the primary antibodies. On the next day, sections were incubated with biotinylated secondary antibodies or Alexa Fluor-conjugated secondary antibodies at room temperature for 1 h. For the detection of biotinylated antibody complexes, sections were incubated in peroxidase-coupled avidin-biotin reagent at room temperature for 1 h (ABC kit, Vector Laboratories, catalog number PK-6100) and were then visualized by DAB (3,3′-diaminobenzidine; Dako, Hamburg, Germany, catalog number K3468). Sections were counterstained with standard hematoxylin or DAPI (4′,6-diamidino-2-phenylindole) to visualize cell nuclei if appropriate. For immunofluorescence stainings of gGFAP/EGFP brains, coronal cryosections (16 µm) were blocked in 5% serum in PBS (see above) for one hour. The primary antibody was incubated for 2 h at room temperature. Sections were then incubated using a red secondary antibody at room temperature for 1 h. Detailed information on primary and secondary antibodies used is given in Table A2.

### 2.7. Digital Image Analysis

Processed sections were digitalized using a Nikon ECLIPSE E200 microscope (Nikon Instruments, Düsseldorf, Germany) equipped with a DigitalSight DS-2Mv camera. Immunofluorescence sections were digitalized using a Leica DMi 8 (Mannheim, Germany) equipped with a Hamamatsu ORCA-Flash 4.0 Digital CMOS Camera (C11440, Hamamatsu, Japan). Sections of hGFAP-EGFP mice were digitalized using a Leica DM 6B (Mannheim, Germany). Quantitative analyses were performed by designing automated image analysis algorithms in Fiji/ImageJ (NIH, Bethesda, MD, USA). The intensity of anti-AQP4 immunoreactivity in the murine and human white matter was evaluated by densitometry. A formula-based threshold (ImageJ threshold ‘IsoData’) was used to automatically identify strongly stained pixels. The optical density was then calculated as the percentage of strongly stained pixels divided by the count of all pixels in the region of interest (ROI). The results were normalized to control group sections (applies for mice tissues) or normal-appearing white matter (NAWM) areas (applies for human tissues).

The following strategy was used to select the ROIs in human post-mortem sections: Serial sections were first processed for anti-PLP and anti-MHC-II (LN3) immunohistochemistry to visualize demyelination and inflammatory activity, respectively. Subsequent slides containing chronic-active lesions [35] were processed for anti-AQP4 immunohistochemistry. Per lesion, two areas containing NAWM and two areas within the center of the lesion were digitalized, and the relative staining intensity quantified as described above. Relative staining intensities from both NAWM and lesion areas per slide were averaged and statistically compared.

For quantification of the density of AQP4-positive microvessels (i.e., capillary density) in the murine cortex, two observers randomly selected one cortical ROI per animal and hemisphere. Two consecutive sections were analyzed. Three control and three cuprizone-intoxicated mice (5 weeks) were included in this analysis. The size of each ROI was measured. The ImageJ operation ‘FindMaxima’ was used to automatically mark strongly stained microvessels, for which the surrounding area is less stained (i.e., local maximum). The capillary density was calculated as the count of local maxima divided by the ROI size. The result was further normalized to control group sections.

Three vessel types were systematically included for the quantification of perivascular anti-AQP4 and anti-GFAP staining intensities in Cup/EAE animals: Type 1 vessels without any inflammatory cells in the perivascular space, Type 2 vessels with an enlarged perivascular space containing peripheral immune cells but trapped behind an intact glia limitans, and Type 3 vessels where inflammatory cells migrated out of the enlarged perivascular space, penetrating the surrounding brain parenchyma. Each vessel was digitalized creating an image containing three spectral layers (DAPI, AQP4, GFAP triple-immunofluorescence). Perivascular immunoreactivity was analyzed in the AQP4 and GFAP layers by using densitometry in concentric rings around the vessel (see Appendix A
Figure A1 for details of the principle). To analyze non-polarized, diffuse staining intensities in single astrocytes, brain section from control and 5 weeks cuprizone intoxicated hGFAP-eGFP mice were processed for anti-AQP4 immunoflourescence stains. To investigate the putative increase in a diffuse astrocytic anti-AQP4 staining intensity, we outlined single astrocyte areas in the green (GFAP) fluorescence channel and measured the relative optical densities in the red (AQP4) fluorescence channel. Optical density analyses were performed using ImageJ, applying the threshold algorithm ‘moments’.

### 2.8. ELISA

The anti-AQP4-IgG titer in blood sera was measured by using an AQP4-ELISA kit (DLD Diagnostika GmbH, Hamburg, Germany) and following the manufacturer’s protocol. Test samples were transferred into a 96-well plate with bound AQP4 and then incubated with biotinylated AQP4 for 2 h (bridge-type ELISA). For the detection of biotinylated antibody complexes, samples were incubated with Streptavidin-peroxidase for 20 min and then visualized by incubation with TMB (3,3′,5,5′-tetramethylbenzidine) for 20 min. The optical density (OD) was measured at 450 nm using a photometer (SpectraMax M3, Molecular Devices, San Jose, CA, USA). The concentration of the samples was calculated from its OD by using the Beer-Lambert law and calibrators provided by the kit. Standardized NMO-patient serum (provided by kit) and a commercially available rabbit anti-murine AQP4 antibody, diluted 1:50 in PBS, (AQP4 H-80 antibody, sc20812, Santa-Cruz, Dallas, TX, USA) were used as positive controls. The lower limit of the test is 0.17 U/mL (data by manufacturer).

### 2.9. Statistical Analysis

Data are given as median + interquartile range (IQR) due to small sample size. Non-Gaussian distribution was assumed. Differences between groups were statistically tested using GraphPad Prism 5 (GraphPad Software Inc., San Diego, CA, USA). To compare two unpaired groups, the Mann-Whitney U test was applied. To compare two paired groups, the Wilcoxon test was applied. To compare two groups using multiple slides from each animal, a repeated-measures ANOVA was applied. To compare more than two groups, the Kruskal–Wallis test followed by Dunn’s post hoc test was applied. *p* values ≤ 0.05 were considered to be statistically significant and are indicated by asterisks: ns = not significant, * = *p* < 0.05, ** = *p* < 0.01, *** = *p* < 0.001, **** = *p* < 0.0001. 

## 3. Results

First, we performed immunohistochemical stains to investigate AQP4 expression and determined whether expression levels are higher in regions that show predominant demyelination in the cuprizone-model. To test the reliability of the applied antibody, we first processed brain sections from control hGFAP-eGFP-mice for anti-AQP4 immunofluorescence staining. As demonstrated in Figure 1A, we found a highly polarized AQP4 expression (red) around small brain vessels, sharply demarcating the glia limitans perivascularis of astrocytes (green). This staining pattern demonstrates that the applied antibody reliably detects AQP4 protein.

Next, we processed brain sections from control and cuprizone-intoxicated mice for chromogen anti-AQP4 immunohistochemistry. In control animals, we found low anti-AQP4 immunoreactivity in the CNS white matter such as in the lateral corpus callosum (Figure 1B,b’), in the medial corpus callosum (Figure 1B,b’’), or the hippocampal fimbria region (Figure 1B star). In the white matter, immunoreactivity was particularly pronounced around small blood vessels and capillaries (arrowheads in Figure 1b’,b’’). In control cortical tissues, cortical microvessels were strongly immunoreactive for AQP4, and the parenchyma exhibited weak, diffuse “background” immunoreactivity (Figure 1B,b’’’). Of note, distinct subcortical grey matter structures, such as the periventricular habenulae (arrow in Figure 1B), showed strong AQP4 immunoreactivity. After 5 weeks cuprizone-intoxication, a robust increase in diffuse anti-AQP4 immunoreactivity was observed in those regions with predominant demyelination and glia cell activation (for example the medial part of the corpus callosum or the hippocampal alveus region; # in Figure 1C). Unbiased quantification of the optical densities in the corpus callosum verified this observation of an increased, diffuse anti-AQP4 immunoreactivity, both in the medial (+271%) and lateral (+252%) parts of the demyelinated corpus callosum (Figure 1D). In contrast, the white matter tracts that are resistant to the cuprizone intoxication, such as the hippocampal fimbria [36], showed a less severe anti-AQP4 immunoreactivity increase. An increase of a diffuse anti-AQP4 immunoreactivity was also observed in analyses, which focused not on the entire corpus callosum but just on areas covered by single astrocytes. These areas were determined in hGFAP-eGFP brain sections processed for anti-AQP4 immunofluorescence (see Figure 1A1). In the demyelinated cortex [37], we observed, on the one hand, an increase in the diffuse “background” immunoreactivity, on the other hand, an obvious loss of the perivascular AQP4 immunoreactivity, indicating loss of AQP4 polarization (Figure 1C,c’’’). To systematically analyze this aspect of AQP4 grey matter pathology, we semi-automatically quantified the numbers of anti-AQP4 immunoreactivity maxima in control- and cuprizone-cortices. As demonstrated in Figure 1E, compared to controls, there was a significant loss of anti-AQP4 immunoreactive capillaries in the cortex of 5 weeks cuprizone-intoxicated mice (−61%).

Next, we asked whether the diffuse AQP4 expression increase is as well observed in post-mortem MS lesions. To this end, seven lesions from seven progressive MS patients were processed for anti-AQP4 immunohistochemistry, and the relative staining intensity was quantified in the NAWM and the lesion center by densitometrical measurements (Figure 2C). As demonstrated in Figure 2A, the anti-AQP4 staining intensity was greater in the center of chronic-active lesions compared to the NAWM. In line with a previous report [38], AQP4 immunoreactivity was even more intense at the border of the lesion than in its center (see arrowheads in Figure 2B).

Our results so far suggest that metabolic oligodendrocyte injury with concomitant demyelination and astro-/microgliosis results in loss of AQP4 polarization paralleled by an increased, diffuse AQP4 expression. In contrast to MS tissues, experimental cuprizone-induced demyelination does not trigger peripheral lymphocyte cell recruitment. Our group recently demonstrated that by combining cuprizone intoxication with active EAE induction, pronounced peripheral immune cell recruitment, including monocytes and lymphocytes into the forebrain occurs [29,30]. We took advantage of this model and investigated AQP4 expression around perivascular, inflammatory lesions. Three vessel types were systematically included in these analyses: Type 1 vessels without any inflammatory cells in the perivascular space, Type 2 vessels with an enlarged perivascular space containing peripheral immune cells but trapped behind an intact glia limitans, and Type 3 vessels where inflammatory cells migrated out of the enlarged perivascular space, penetrating the surrounding brain parenchyma. Around normal type 1 vessels, as expected, we found strong anti-AQP4 immunoreactivity, and AQP4 expression levels decreased with increasing distance from the vessel center (Figure 3A left column, and Figure 3B black line). Around type 2 vessels, anti-AQP4 immunoreactivity was weaker at close vicinity to the vessel but remained constant with increasing distance from the vessel center (Figure 3A center column, and Figure 3B grey line). Of note, around type 3 vessels, we found very weak perivascular anti-AQP4 immunoreactivity, but AQP4 expression levels increased with increasing distance from the vessel center (Figure 3A right column, and Figure 3B dotted line). In line with this observation, the relative anti-AQP4 staining intensity close to the vessel center was highest around Type 1 (1.55 AU), and lowest around Type 3 vessels (0.69 AU; see Figure 3D). This unbiased, objective evaluation suggests that peripheral immune cell recruitment results in a loss of anti-AQP4 staining intensity. The same staining pattern was not observed for GFAP (Figure 3C,E). On the contrary, anti-GFAP staining intensity was increased around type 2 compared to type 1 vessels (see Figure 3E), suggesting that loss of AQP4 staining intensity is not due to entire destruction of perivascular astrocytes.

Loss of astrocytic AQP4 expression is a well-described histopathological characteristic of NMO [12,39], and antibodies against AQP4 have been reported to mediate NMO pathology [40]. We were thus interested whether, in our experimental mice, anti-AQP4 antibodies can be detected. To this end, the serum of control, EAE, cuprizone, and Cup/EAE mice was sampled and the presence of anti-AQP4 antibodies in the serum was analyzed via ELISA. As demonstrated in Figure 4, high titers of anti-AQP4 antibodies were detected in serum samples from NMO patients (i.e., positive control of the applied assay). Furthermore, high titers were detected in serum samples that were supplemented with a commercially available rabbit anti-murine AQP4 antibody. In contrast, we were not able to detect anti-AQP4 antibodies in the serum of either control, EAE, cuprizone, or Cup/EAE mice.

It has been shown that abundant neutrophil granulocytes are present in a subset of active NMO lesions [41]. To address this aspect, we processed inflammatory lesions of Cup/EAE mice for anti-granulocyte immunohistochemistry. As demonstrated in Figure 4B, neutrophil granulocyte numbers were virtually absent in the brains of control mice, but numerous were found in Cup/EAE mice. In human brain and spinal cord tissue, astrocytes express (besides AQP4) another water channel, i.e., aquaporin 1 (AQP1). This is different from rodent CNS tissue, where AQP1 is exclusively expressed on the epithelial cells of the choroid plexus [12,42]. In some NMO lesions (lesion type 3, according to Misu et al.) densely packed GFAP and AQP1 reactive astrocytic processes have been described [12]. To investigate whether AQP1 expression is induced in our model, brain sections from Cup/EAE mice were processed for anti-AQP1 immunohistochemistry. As demonstrated in Figure 4B (bottom row), in both, control and Cup/EAE mice, strong AQP1-expression was found in the epithelial cells of the choroid plexus, but not in the brain parenchyma.

## 4. Discussion

In this work, we demonstrated a pathologic AQP4 expression profile in a metabolic and inflammatory MS animal model. After metabolic, cuprizone-induced demyelination, we found a robust increase in diffuse anti-AQP4 immunoreactivity paralleled by a loss of the polarized, perivascular AQP4 expression. The diffuse AQP4 expression increase was verified in chronic-active lesions of progressive MS patients. Beyond, we found an intense loss of AQP4 staining intensity around inflammatory lesions in Cup/EAE mice. This was paralleled by the recruitment of neutrophil granulocytes, but an absence of circulating anti-AQP4 antibodies and AQP1-expression induction (Figure 5).

AQP4 is reported to be the most abundant water channel in the brain, spinal cord, and optic nerve, and is primarily localized to specific cellular subregions, such as astrocyte foot processes facing either the periventricular space, vessels or the brain surface [17]. In line with this, we found that astrocytes exhibit highly differentiated AQP4 immunolabeling with the predominant signal concentrated in glial processes close to or in direct contact with blood vessels (see Figure 1A,b’’’). As previously demonstrated, neurons throughout the cortex were consistently negative [17]. In line with another report, we found a strong anti-AQP4 immunoreactivity in the habenula region, lining the third ventricle [43], further demonstrating that the applied antibody is specific.

The pattern of AQP4 expression at the borders between the brain and major water-containing compartments suggests that AQP4 facilitates the flow of water into and out of the brain. *Aqp4* deletion is associated with a sevenfold reduction in cell plasma membrane water permeability in cultured astrocytes [20], but *Aqp4* null mice have normal intracranial pressure and only slightly increased total brain water content [22,44]. These findings suggest that AQP4 is not required for relatively slow water movements into and out of the brain that take place under normal physiological conditions, but might be relevant during brain pathologies, where the rates of water flow into and out of the brain are significantly increased. In line with this, there is less edema formation in *Aqp4* knockout compared to wild-type mice after bacterial meningitis [22], focal cerebral ischemia [19], water intoxication [44], or spinal cord injury [45]. In models of disease involving blood–brain barrier disruption, including brain tumor [22], brain abscess [46], subarachnoid hemorrhage [47], and status epilepticus [48], *Aqp4* null mice develop more severe brain edema than wild-type mice, suggesting that the mode of edema formation (vasogenic versus cytotoxic edema) is important to define the role of AQP4. Edema formation, to a variable extent, has been demonstrated in MS [49,50,51,52] as well as the EAE [53,54] and cuprizone animal models [55,56]. Comparable to our results, Wolburg-Buchholz and colleagues demonstrated the loss of polarized AQP4 localization in astrocytic end-feet surrounding microvessels during EAE [57]. In particular, the authors found a constant increase of anti-AQP4-related fluorescence intensity in the areas of perivascular inflammatory cuffs correlating to the severity of EAE, which was paralleled by a redistribution of AQP4 over the astrocytic surface detected by immunogold labeling with subsequent electron microscopy analysis. Of note, western blot analysis failed to detect enhanced AQP4 protein levels, suggesting that the observed diffuse increase in anti-AQP4 labeling is at least in part due to a redistribution phenomenon. Unpublished data from our group showed that *Aqp4* mRNA expression levels are increased in the corpus callosum and cortex of mice intoxicated with cuprizone for 5 weeks. This increase, however, was modest and did not explain the strong increase in the diffuse staining intensity. A just modest *Aqp4*-mRNA increase has also been reported in the optic nerves [58] and spinal cords [59] of mice with EAE. Beyond, reactive astrocytes isolated from the brains of mice treated either with lipopolysaccharide or subjected to experimental stroke showed no increase in *Aqp4*-mRNA expression levels [60]. The latter finding, however, might be because astrocytes in culture display an activated phenotype. In summary, these results suggest that the observed increase in the diffuse anti-AQP4 immunoreactivity is due to AQP4 redistribution rather than increased protein expression. Increased diffuse anti-AQP4 staining and moderately increased *Aqp4*-expression have also been reported by others in the cuprizone model [55]. Although we were able to reproduce the findings of Berghoff et al. showing a diffuse increase of anti-AQP4 staining intensity in the corpus callosum during cuprizone-induced demyelination, we cannot fully confirm their previous findings that the anti-AQP4 staining pattern remains unchanged in the demyelinated cortex. Our present study revealed a significant loss of polarized, perivascular AQP4 expression. Whether polarization loss also occurs in the corpus callosum is likely, but hard to quantify due to the strong increase in the diffuse anti-AQP4 staining intensity. On the functional level, the EAE disease course is greatly attenuated in *Aqp4* knockout mice [61]. Whereas most wildtype mice developed progressive tail and hindlimb paralysis, clinical signs were virtually absent in *Aqp4* null mice suggesting that the reduction in AQP4 water transport may be protective in neuroinflammatory CNS diseases [62]. Further studies will be key to show whether AQP4 deficiency influences disease severity in either cuprizone or Cup/EAE mice.

In Cup/EAE mice, we observed, despite a diffuse increase in anti-AQP4 staining intensity (data not shown), a perivascular loss of anti-AQP4 immunoreactivity. While this is the first work investigating the expression of AQP4 in a combined metabolic (i.e., cuprizone) + autoimmune (i.e., EAE) MS animal model, a decreased expression of AQP4 within and around inflammatory lesions has been described in EAE spinal cord tissues [63,64]. Of note, a comparable patchy diminution of anti-AQP4 immunoreactivity has also been observed in an autopsy case of the Marburg variant of MS [65]. To investigate whether AQP4 autoantibodies induce the reduced AQP4 expression around the inflammatory lesions, the production/presence of AQP4 antibodies was investigated in the experimental mice. However, we found no evidence for endogenous AQP4 antibody production in either cuprizone, EAE, or Cup/EAE mice. The presence of AQP4 antibodies in EAE is discussed controversially. While one study found no serum AQP4 antibodies [62], another study reported that levels were increased, and correlated positively with the clinical score [66]. While the applied EAE model was comparable in both studies (i.e., MOG_35–55_-induced EAE in C57BL/6J mice), the methods used to detect AQP4 antibodies in the serum differed. In the latter study, serum mouse AQP4-Ab levels were measured by ELISA, while a cell-based assay was used by Li and colleagues. In line with the study of Li and colleagues, we did not find serum AQP4-Ab in either experimental or control groups.

The relevance of our in vivo findings was verified by analyzing the expression pattern of AQP4 in post-mortem MS samples. In our progressive MS patient cohort, we found a diffuse increase of the anti-AQP4 immunoreactivity in the center of chronic-active lesions compared to the NAWM. Our findings are in line with previous reports demonstrating increased AQP4 expression in MS lesions [13,67]. In this study, the authors conducted a semi-quantitative microscopic analysis of AQP4 expression. AQP4 expression was graded as absent, present, moderate, or abundant. Of note, the highest levels of AQP4 expression were observed in areas of chronic active lesions. Enhanced expression of AQP4 has also been demonstrated in human brains with infarction [68], in various inflammatory conditions including human immunodeficiency virus encephalitis and progressive multifocal leukoencephalopathy [38] and some brain tumor series [69,70]. These results suggest that the diffuse increase in anti-AQP4 immunoreactivity is not specific to MS but rather reflects astrocyte pathology in a broader context.

Several similarities exist between NMO lesions and the lesions observed in Cup/EAE mice. For example, both lesions are characterized by AQP4 loss (demonstrated in this study), a variable extent of demyelination and oligodendrocyte loss, axonal injury, and neutrophil recruitment [12,29,30,71]. Specific features in acute stages of NMO lesions are, besides the loss of AQP4, perivascular or subpial deposition of humoral factors such as immunoglobulin IgG and IgM or activated complement. In contrast, CD3^+^ and CD8^+^ T cells are rare in such lesions [39,41]. This apparently reflects a humoral immune attack against the glia limitans, especially against the astrocyte foot processes by AQP4 reactive autoantibodies. Although we did not find endogenous AQP4-Ab production in our mice, one should keep in mind that a significant percentage of NMO patients are seronegative [72,73]. Beyond, the liquor might contain AQP4 reactive autoantibodies, which was not tested in the current study. Further studies are, thus, needed to address this aspect of the Cup/EAE model.

In summary, we provide evidence that the diffuse increase in anti-AQP4 staining intensity is due to a metabolic injury to the brain, whereas the focal, perivascular loss of anti-AQP4 immunoreactivity is mediated by peripheral immune cells. Future studies are now needed to understand the mechanism operant during Cup/EAE lesion development and to relate these mechanisms to NMO and MS.

## Figures and Tables

**Figure 1 cells-09-02187-f001:**
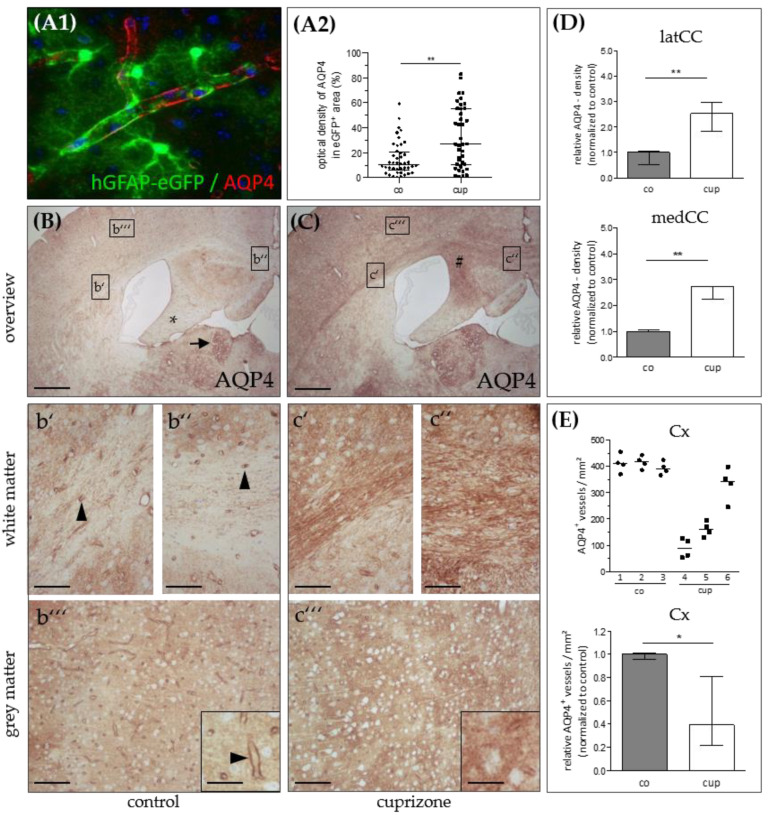
Aquaporin-4 (AQP4) expression in the cuprizone model. (**A1**) Representative image demonstrating expression of AQP4 (red) in eGFP-expressing astrocytes (green). (**A2**) Relative anti-AQP4 staining intensities (control and 5 weeks cuprizone) within the area covered by single eGFP-expressing astrocytes. *Data are shown as median + IQR; tested* via *Mann-Whitney U; co n = 44, cup n= 39.* (**B**) AQP4 expression in control (co) animals. The lateral corpus callosum (**b’**), medial corpus callosum (**b’’**), and grey matter cortex (**b’’’**) are shown in higher magnification. The star highlights the hippocampal fimbria region; the arrow highlights the habenula; the arrowheads highlight AQP4-immunoreactive capillaries. (**C**) AQP4 expression in animals intoxicated with cuprizone for 5 weeks. The lateral corpus callosum (**c’**), medial corpus callosum (**c’’**), and grey matter cortex (**c’’’**) are shown in higher magnification. The hash sign highlights the hippocampal alveus. *Scales: (**B**,**C**) 180 µm, (**b’**–**b’’’**,**c’**–**c’’’**) 90 µm, inserts 30 µm.* (**D**) Semi-automatic optical density analyses of anti-AQP4 immunoreactivity in the lateral (latCC) and medial (medCC) corpus callosum in control mice and after 5 weeks cuprizone intoxication. (**E**) Semi-automatic quantification of anti-AQP4-positive microvessel densities in the cortex (Cx) per mm² in control mice and after 5 weeks cuprizone intoxication (top: absolute values, bottom: relative values). *Data are shown as median + IQR; tested* via *repeated measures ANOVA; n = 3 per group; * = p < 0.05, ** = p < 0.01.*

**Figure 2 cells-09-02187-f002:**
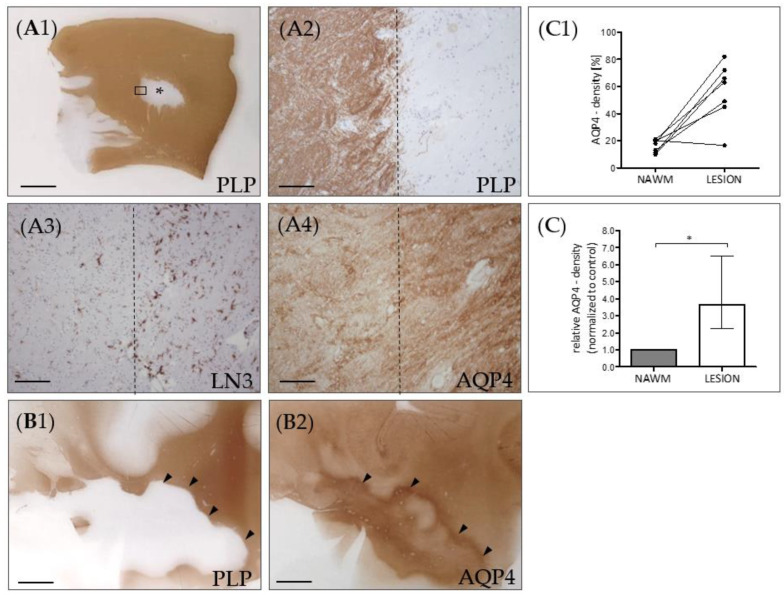
AQP4 expression in chronic-active multiple sclerosis (MS) lesions. (**A**) Representative MS lesion stained for anti-PLP. The middle part of the image shows a focal white matter demyelinated lesion (star). The area highlighted by the rectangular box in (**A1**) highlights the border of the white matter lesion and is shown in (**A2**) in higher magnification. Dotted lines demarcate the border of the lesion. (**A3**) shows the lesion border of a consecutive section, processed for anti-MHC class II (LN3) immunohistochemistry to label activated microglia and macrophages. (**A4**) shows the lesion border, processed for anti-AQP4 staining. (**B**) Another representative lesion processed for anti-PLP (**B1**) and anti-AQP4 (**B2**) immunohistochemistry. The arrowheads highlight the border of the lesion. Note the highest AQP4 expression levels at the lesion border. *Scales: (**A1**,**B1**,**B2**) 3 mm, (**A2**–**A4**) 180 µm.* (**C**) Semi-automated optical density analyses of anti-AQP4 immunoreactivity in demyelinated lesions and normal appearing white matter (NAWM) of the same section. Absolute values are shown in the top row (**C1**), relative values are shown in the bottom row (**C2**). *Data are shown as median + IQR and tested* via *Wilcoxon-test; n = 7 per group; ns = not significant, * = p < 0.05.*

**Figure 3 cells-09-02187-f003:**
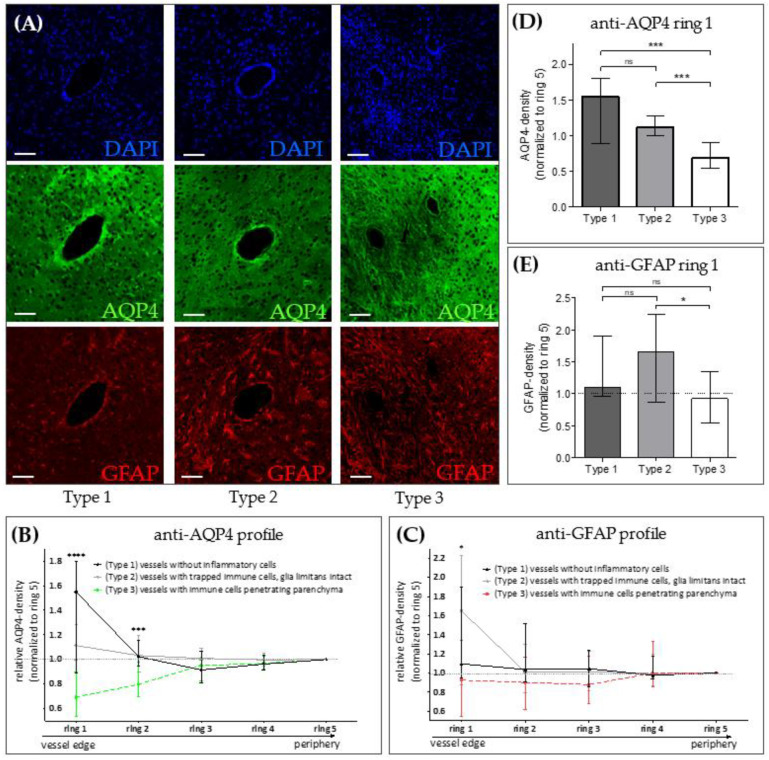
AQP4 expression in the Cuprizone/EAE model. (**A**) Representative images of forebrain vessels in Cuprizone/EAE mice, shown in three spectral layers (triple-immunofluorescence): DAPI (blue, upper row), AQP4 (green, middle row), GFAP (red, bottom row). Left column: Type 1 vessels (no inflammatory cells in the perivascular space). Middle column: Type 2 vessels (immune cells trapped within an intact glia limitans). Right column: Type 3 vessels (immune cells penetrating the brain parenchyma). *Scales: 30 µm.* (**B**,**C**) Semi-automated evaluation of perivascular optical densities in AQP4 and GFAP spectral layers in 5 concentric rings around the vessel (see Appendix A
Figure A1 for details of the applied evaluation method). Ring 1 represents the vessel’s edge, ring 5 its outermost peripheral part. (**D**,**E**) Comparison of anti-AQP4 and anti-GFAP optical densities in ring 1. *Data is shown as median + IQR and was tested* via *Kruskal-Wallis test (**B**,**C**) followed by Dunn’s post-hoc test for ring 1 (**D**,**E**); type 1 n = 9; type 2 n = 24, type 3 n = 22; ns = not significant, * = p < 0.05, *** = p < 0.001, **** = p < 0.0001.*

**Figure 4 cells-09-02187-f004:**
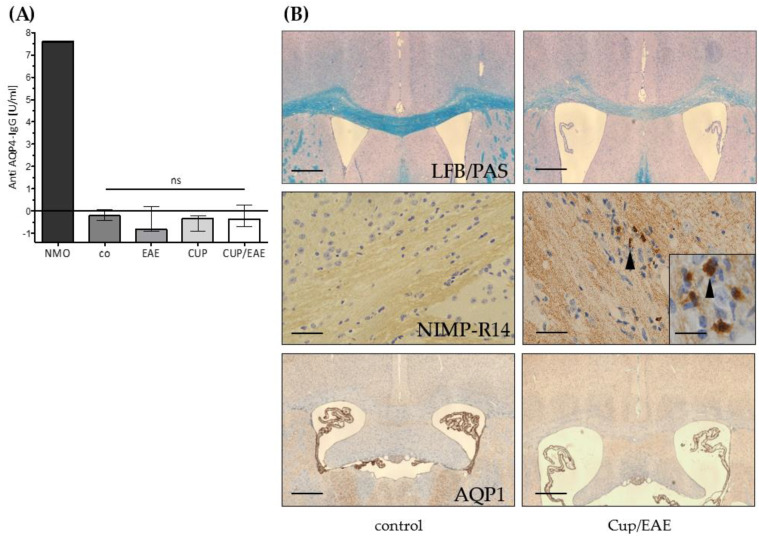
Anti AQP4-IgG serum levels. (**A**) Anti- AQP4 IgG levels in blood sera of the control group (co) and after EAE, cuprizone (CUP), or Cuprizone/EAE (CUP/EAE) induction. Neuromyelitis optica (NMO)-patient serum (provided by kit) was used as positive control. Commercial anti-mouse AQP4-antibody (sc20812) 1:50 diluted in PBS was tested positive with 45.34 U/mL (not shown). *Data is shown as median + IQR and tested* via *Kruskal-Wallis test; co n = 5, CUP n = 5, EAE n = 10, CUP/EAE n = 10.* (**B**) Representative images of control and Cup/EAE brain sections, processed for either LFB/PAS (top row), anti-NIMP-R14 immunohistochemistry (middle row), or anti-AQP1 immunohistochemistry (bottom row). Arrowheads in the middle row indicate granulocytes. *Scales: top & bottom 180 µm; middle: 90 µm; insert 30 µm.*

**Figure 5 cells-09-02187-f005:**
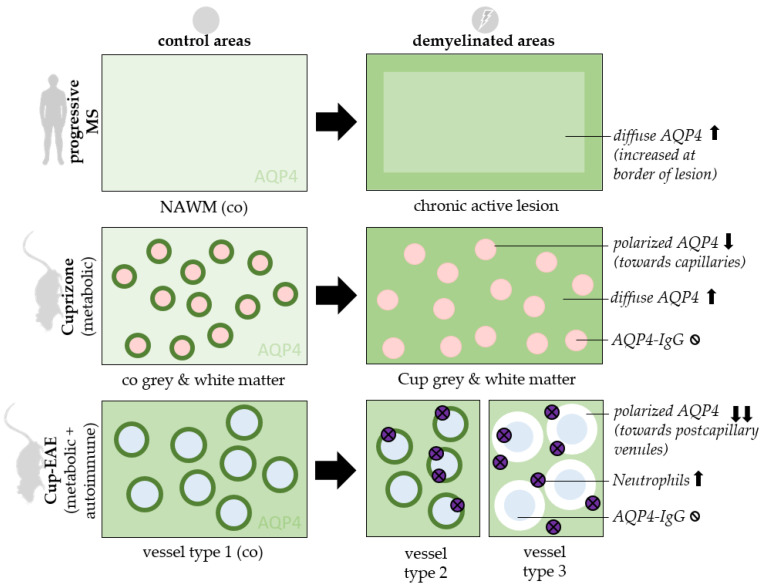
Synopsis of findings. AQP4 staining patterns are displayed in control areas (**left**) and demyelinated areas (**right**). 3 experimental settings are shown: Chronic active lesions of progressive MS patients (**top row**), mice treated with 5 weeks Cuprizone intoxication (**middle row**) and mice treated with Cuprizone + EAE (**bottom row**). *Ø = absent, ↑ = increased, ↓ = reduced, co = control, cup = cuprizone, NAWM = normal appearing white matter, red circles = capillaries, blue circles = postcapillary venules.*

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
