# Peer review of "Aquaporin-4 Expression during Toxic and Autoimmune Demyelination"

_cells, 2020, doi:10.3390/cells9102187_

Round 1

Reviewer 1 Report

Review comments for cells-900511     

The authors claim that one of the water channel AQP4 expression was altered and diffusible in the brain with cuprizone mediated demyelinated mouse due to a metabolic brain injury.

Their aim is interesting but still needs some additional experiments to publish the paper.

1) In Figure 1, the authors need to present a data for hGFAP-eGFP/AQP4 mice with cuprizone in parallel to (A). Because AQP4 expression would be not only astrocyte endfeet but also dispersed to its cell body, the difference should be presented. 

Thus, only (C) is not enough to detect AQP4 expression in entire astrocyte cell body.

2) In which type of cells expressed PLP and AQP4 in (B1) and (B2), respectively in Figure 2?

Also, what cells are lining at the lesion border?

What is the border composed of?

3) In Figure 3, the authors need to analyse existence of immune cells around vessels using some of the markers such as anti CD45 antibody or others for them. Furthermore, they need merged figure of AQP4 and GFAP expression to know the diffusible AQP4 expression in astrocytes.

4)In Figure 4, the graph must be “(A)” so it should be clarified.

In figure 4(B), what the authors used to identify granulocytes? Antibody against the cells? They need to mention it.

Also, they need higher magnification for middle panel of (B) to clarify the cell shape.

Furthermore, the letter in bottom row panel must be “AQP4”.

5) In the Discussion section, the authors wrote as an AQP4 review paper. However, the research paper should include further experimental discussion.

For example, ….

(a) AQP4 expression was expanded in which type of cells?

(b) How did granulocytes enter into the brain parenchyma other than macrophage or T/B-cells ?

Has BBB broken down with cuprizone?

(c) How the demyelination inducing cuprizone cause diffuse AQP4 expression?

6) In the Introduction section, the authors must summarize the information about cuprizone because I suspect little is understood about it for the most of the readers of this paper.  

Reviewer 2 Report

The paper describes an important role of AQP4 in toxic and autoimmune  demyelinization, with potential importance in for instance multiple sclerorosis (MS). The presentation of the results are clear and straight-forward to interprete, as are the conclusions.

The authors are still recommended to:1. Have the writing reexamined for instance regarding use of pronouns instead of substantives repeatedly througout the text; 2. Make a more clear summary of the present findings and importance in relation to previously published results; 3. Present a visual model of the findings in relation to the anticipated demyelinization effects. 4. Describe more clearly why cuprizone (Cup) is the most relevant toxic substance to be employed in the EAE model; 5. Focus and shorten the Introduction and the Discussion with relevance for the present study and the findings therein.

Author Response

Please find the attachment below.

Round 2

Reviewer 1 Report

I just satisfied with the authors extensive revision.

So, now the manuscript and the data became high level to publish in Cells.

Please include all the revised data in revised version.